# MechVQA: Benchmarking and Enhancing Multimodal LLMs on Comprehensive Mechanical Drawing Understanding

Qian Kou [* 1]  Xiaofeng Shi [* 1]  Yulin Li [1 2]  Xiaosong Qiu [1]  Xinyang Wang [3]  Hua Zhou [1]  Cao Dongxing [3]

## Abstract

Multimodal Large Language Models (MLLMs) have demonstrated significant achievements in general visual question answering (VQA) tasks. However, they remain brittle on mechanical engineering drawings, where high annotation density and weak domain knowledge, compounded by unreliable spatial relation reasoning under strict projection rules and geometric constraints, make decisive cues easy to miss and frequently lead to wrong answers. To bridge this gap, we introduce the first comprehensive mechanical drawing understanding dataset, **MechVQA**, created through a semi-automated construction and quality-control pipeline. **MechVQA** contains 3.3k high-density pictures with 21K question–answer pairs, spanning 10 different fine-grained tasks across three capability levels: **Recognition**, **Reasoning**, and **Judging**, providing a testbed to evaluate and improve MLLM understanding on real-world mechanical drawings. On top of **MechVQA**, we then develop the **MechVL** model through a multi-stage training paradigm, building a strong domain-specialized baseline. Extensive experimental results demonstrate that **MechVL** outperforms the strongest closed-source baseline by 7.57 percentage points on the MechVQA total score, significantly enhancing mechanical drawing understanding ability and providing a reusable foundation for deploying MLLMs in mechanical design and inspection scenarios.

## 1. Introduction

Mechanical engineering drawings are the primary medium for communicating geometry, tolerances, and assembly intent in mechanical design and inspection. Unlike natural images, a drawing encodes semantics through a compact, standardized graphical language that combines orthographic multi-view projections, dense dimensioning, section views, symbolic notations, and structured textual content. As illustrated in Figure 1(a), understanding such drawings requires more than generic visual perception: a model must (i) recognize high-density dimensions, callouts, and domain-specific symbols, (ii) reason about spatial relations under projection rules via cross-view feature correspondence, and (iii) apply drafting standards to interpret conventions such as geometric tolerances.

Despite rapid progress in Multimodal Large Language Models (MLLMs), general-purpose models remain brittle on mechanical drawings (Bai et al., 2025; Zhang et al., 2024; Suzuki & Matsuo, 2022). The dominant failure mode is not merely optical recognition. High annotation density and symbol clutter make decisive cues easy to miss, and weak domain priors with unreliable spatial relation reasoning under strict projection rules and geometric constraints often yield structurally inconsistent interpretations and incorrect answers. Similar gaps between foundation models and expert-level performance have been observed across scientific and engineering domains, motivating domain-centric benchmarks that probe capabilities beyond generic VQA. (Burgess et al., 2025; Li et al., 2025)

There remains a lack of dedicated data that systematically covers comprehensive mechanical drawing understanding. Existing multimodal benchmarks in adjacent engineering settings are complementary but scoped to specific slices, such as rulebook-grounded requirements QA, blueprint symbol recognition, or AEC floor-plan literacy, and engineering drawing analysis is often only a small, non-open component within broader suites (Doris et al., 2024; Shteriyanov et al., 2025; Kondratenko et al., 2026; Picard et al., 2025). Consequently, they do not provide a unified evaluation of mechanical part and complex assembly drawings that jointly stresses structured perception, multi-view consistency, and engineering-grade reasoning.

---

[*]Equal contribution [1]Beijing Academy of Artificial Intelligence (BAAI), China [2]Institute of Information Engineering, Chinese Academy of Sciences, China [3]Beijing University of Technology, China. Correspondence to: Qian Kou <kouqian@baai.ac.cn>, Xiaofeng Shi <xfshi@baai.ac.cn>.

*Proceedings of the 43rd International Conference on Machine Learning*, Seoul, South Korea. PMLR 306, 2026. Copyright 2026 by the author(s).

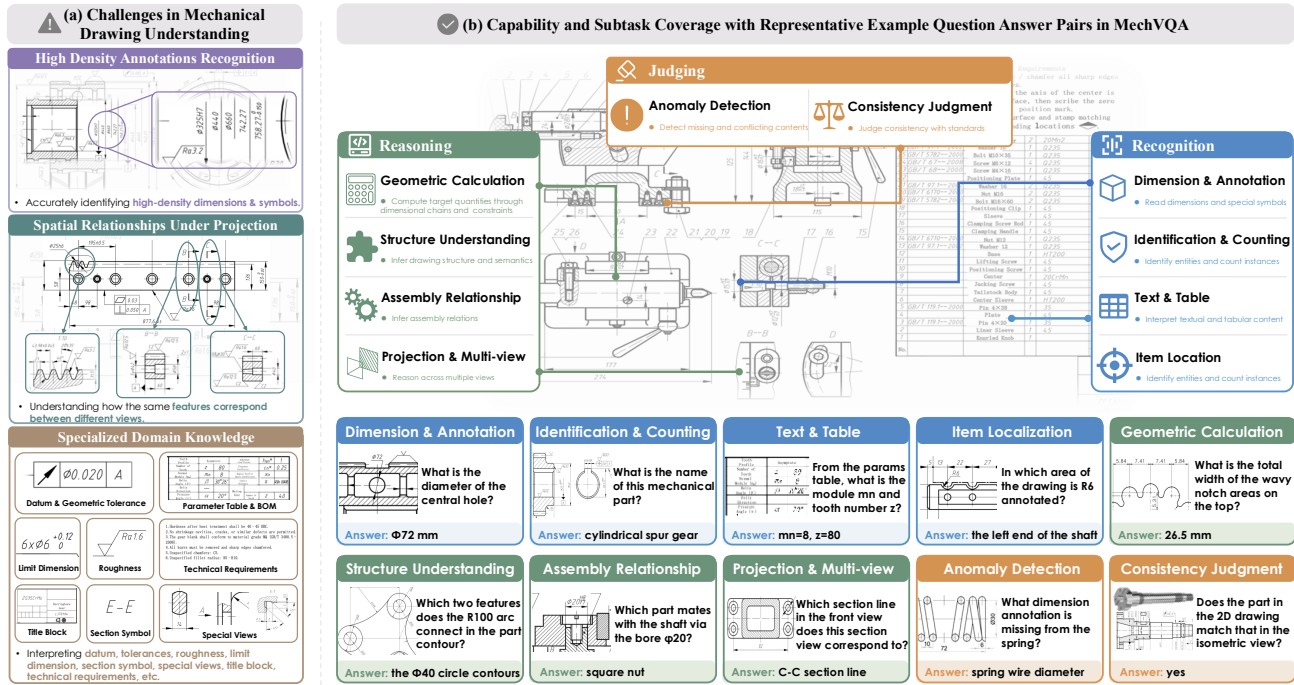

*Figure 1.* **Overview of Mechanical Drawing Understanding and MechVQA.** (a) **Representative challenges**, including high-density annotation recognition, projection-consistent spatial correspondence across views, and standards-aware interpretation of domain-specific symbols, specifications, and tables. (b) **MechVQA task taxonomy** organized into three capabilities (*Recognition, Reasoning, Judging*) with fine-grained subtasks, accompanied by representative question–answer examples grounded in mechanical drawings.

In this work, we study *mechanical drawing understanding with MLLMs* under orthodox drafting conventions, focusing on real part and complex assembly drawings collected from publicly available textbooks, professional handbooks, and design platforms. We introduce **MechVQA**, a benchmark containing 3.3K high-quality drawings and 21K question-answer pairs. MechVQA is organized into three capability axes—*Recognition*, *Reasoning*, and *Judging*—and further decomposed into 10 fine-grained subtasks that jointly cover the core aspects of mechanical drawing understanding. Questions are also stratified into three difficulty levels, enabling systematic evaluation from direct perception to expert-level inference. Figure 1(b) presents representative examples for all subtasks, and Table 1 compares MechVQA with general and CAD/mechanical VQA benchmarks.

Building on MechVQA, we establish **MechVL**, a strong domain-specialized baseline trained through a multi-stage post-training pipeline. We first perform supervised instruction tuning (SFT) to obtain a reference policy for mechanical drawing understanding, and then further improve reliability and task performance through reinforcement learning with **DAPO** (Yu et al., 2025). A key design is a taxonomy-aligned reward scheme that directly optimizes answer correctness, output format compliance, and explanation quality. Concretely, we combine three complementary rewards: (i) a *format reward* to enforce strict output structure and schema va-

lidity, (ii) an *accuracy reward* for factual correctness against ground-truth answers with numeric- and unit-sensitive handling, and (iii) a *quality reward* from an LLM-as-a-Judge that evaluates responses along the axes of *Logic*, *Normativity*, and *Professionalism* (Bai et al., 2024; Gu et al., 2024). This design directly targets common failure cases of SFT-only baselines on dense mechanical drawings, including missed decisive annotations, violated multi-view consistency, and numerically plausible but constraint-inconsistent results.

Across extensive experiments, MechVL achieves consistent improvements over strong general-purpose MLLMs and surpasses closed-source baselines by 6% on the MechVQA total score, providing a reusable foundation for drawing-centric mechanical design and inspection workflows. The main contributions of this work are summarized as follows:

- We introduce **MechVQA**, a benchmark for mechanical drawing understanding built from real part and assembly drawings, organized into three capability axes—*Recognition*, *Reasoning*, and *Judging*—with 10 fine-grained subtasks and three difficulty levels.

- We establish **MechVL**, a domain-specialized baseline trained with multi-stage post-training, including supervised fine-tuning and DAPO-based self-play reinforcement learning with taxonomy-aligned rewards.

- We provide extensive benchmarking and ablation results showing that domain-specialized post-training substantially improves performance on dense mechanical drawings, especially for cross-view reasoning, constraint-sensitive inference, and standards-aware judgment.

## 2. Related Work

**MLLMs and Visual Question Answering.** Pre-trained on large-scale multimodal corpora, Multimodal LLMs excel in general tasks like image–text retrieval and VQA. Visual Instruction Tuning further enhances their instruction-following capabilities, as demonstrated by LLaVA, MiniGPT-4, and Gemini 1.5 (Liu et al., 2023; Zhu et al., 2023; Google, 2024). However, general-purpose models often lack domain-specific drafting knowledge (e.g., projection conventions and tolerances), causing misinterpretations in engineering analysis.

Traditional VQA benchmarks focus on everyday scenes (Antol et al., 2015), leaving a gap in engineering and CAD domains. Recent efforts have begun addressing this: CReFT-CAD (Niu et al., 2025a) explores three-view reasoning, PHT-CAD (Niu et al., 2025b) focuses on parametric primitive analysis. While mechanical reasoning benchmarks like MechBench (Rodríguez et al., 2025) probe physical laws via schematic puzzles, they lack text-rich engineering drawing contexts. DesignQA (Doris et al., 2024) integrates professional documentation, while recent studies on 2D engineering drawings further explore annotation parsing and structured information extraction (Khan et al., 2026). However, a unified benchmark jointly evaluating recognition, reasoning, and judgment on complex mechanical drawings remains absent.

**Instruction tuning with SFT and RL.** A prevailing post-training paradigm first learns instruction following through SFT and then enhances performance via RL (Ouyang et al., 2022; Han et al., 2023; Rozière et al., 2023). Recent advances like DeepSeek-R1 demonstrate that GRPO (Guo et al., 2025) can significantly boost reasoning capabilities by estimating advantages from group-normalized rewards, eliminating the memory-intensive value critic. To better handle structured outputs, GSPO (Zheng C, 2025) further refines this line of work by optimizing for sequential consistency and logical integrity across sampled response paths. Building on the group-based optimization family, DAPO (Yu et al., 2025) retains the group-normalized advantage estimator while introducing asymmetric clipping, dynamic sampling, token-level policy gradients, and overlong reward shaping, improving stability and sample efficiency for long-form reasoning. This is particularly relevant to mechanical engineering, where dimensioning and assembly analysis often require reliable multi-step reasoning.

## 3. MechVQA Dataset

This section describes our data construction pipeline with expert-oriented quality control, as illustrated in Figure 2a. The MechVQA dataset is tailored to practical mechanical engineering needs and is guided by three principles: diversity, professionalism and evaluation effectiveness.

### 3.1. Data Sources and Collection

To support both training and evaluation of MLLMs for mechanical drawing understanding, we curate drawings from publicly available high-quality sources, including mechanical textbooks, professional handbooks, and design platforms. These drawings span a broad spectrum, covering conventional 2D orthographic sheets and drawings augmented with isometric views, as well as both part and assembly drawings, thereby reflecting diverse representative mechanical design scenarios. Since the benchmark is constructed from public educational and professional sources rather than proprietary industrial archives, we view MechVQA as a benchmark for standards-oriented public mechanical drawings, while recognizing that legacy blueprints and company-specific drafting practices remain outside the current scope.

We adopt an efficient yet rigorous preprocessing pipeline to ensure data quality and controllability. Low-quality, incomplete, and poorly scanned drawings are first removed by domain experts, resulting in a corpus of 3,281 high-quality drawing images. We then apply an advanced OCR model (Niu et al., 2025c) to extract textual content (e.g., table entries) and leverage strong closed-source MLLMs (OpenAI, 2025; Google DeepMind, 2025a; Anthropic, 2025b) to infer other raw metadata fields. Finally, mechanically trained graduate students conduct secondary verification under an internal annotation handbook covering checked fields, view-category definitions, workflow, and uncertain-case notes. These verified annotations form the basis for downstream question-answer pair construction; Appendix B.2 reports a correction-rate analysis of expert verification. Examples of mechanical drawings and metadata schema details are provided in Appendix B.1.

### 3.2. Capability and Task Design

To systematically probe MLLMs' capability on mechanical drawings and lay a principled foundation for downstream question construction and difficulty stratification, we organize questions into three capability axes, **Recognition**, **Reasoning**, and **Judging**, and further decompose them into ten subtasks:

- **Recognition** focuses on extracting and grounding explicit information, including text, dimensions, annotations, symbols, and view or region references. This capability includes four subtasks: ***Identification &***

| Datasets | Images | Questions | CAD/Mech | 3D views | Task Focus / Features |
|---|---|---|---|---|---|
| VQA (Antol et al., 2015) | 250,000 | 750,000 | × | × | Gen. VL Grounding |
| COCO-QA (Ren et al., 2015) | 123,287 | 117,684 | × | × | Basic Object/Color Recognition |
| MMBench (Liu et al., 2024) | 2,974 | 2,974 | × | × | Comp. Reasoning Eval. |
| VaseVQA (Ge et al., 2025) | 31,773 | 93,544 | × | × | 2D Archaeology Analysis |
| VaseVQA-3D (Zhang et al., 2025) | 664 | 4,460 | × | ✓ | 3D Archaeology Analysis |
| AECV-Bench (Kondratenko et al., 2026) | 120 | 192 | ✓ | × | AEC Drawing & Spatial OCR |
| MechBench (Rodríguez et al., 2025) | - | - | ✓ | × | Physics & Mech. Components |
| DesignQA (Doris et al., 2024) | - | 1,451 | ✓ | ✓ | CAD Design Compliance |
| TriView2CAD (Niu et al., 2025a) | 609,000 | 160,000 | ✓ | ✓ | Ortho-Projection Reasoning |
| FCM (Picard et al., 2025) | 378 | 1,000+ | ✓ | ✓ | Design to Manufacturing |
| BlueprintSymVL (Shteriyanov et al., 2025) | 200 | 200 | ✓ | × | Blueprint Symbol Recognition |
| MechVQA(Ours) | 3,281 | 20,778 | ✓ | ✓ | Mechanical Drawing Understanding |

*Table 1.* Comparison of VQA Datasets: General vs. CAD/Mechanical Domains.

*Counting (IC)*, *Dimension & Annotation (DA)*, *Text & Table (TT)*, and *Item Localization (IL)*.

- **Reasoning** targets multi-step inference beyond direct reading, such as geometric calculation, cross-view projection constraints, and compositional understanding of structures and assemblies. This capability contains four subtasks: *Structure Understanding (SU)*, *Geometric Calculation (GC)*, *Assembly Relationship (AR)*, and *Projection & Multi-view (PM)*.

- **Judging** centers on decision-making under engineering rules, requiring models to detect anomalies and verify consistency or standards compliance. It contains two subtasks: *Anomaly Detection (AD)* and *Consistency Judgment (CJ)*.

Each question is assigned exactly one subtask label. This design encourages models to align visual elements with mechanical terminology, integrate domain knowledge, and perform multi-step reasoning and verification, rather than merely detecting lines or text. Full subtask and difficulty-level definitions are provided in Appendix B.3.

### 3.3. Question and Answer Generation

Starting from the curated drawing images and the expert-verified metadata obtained in Section 3.1, we generate MechVQA question-answer pairs following the capability hierarchy and subtask taxonomy introduced in Section 3.2.

**Overview.** Given a drawing and its metadata, we first generate candidate questions that are grounded in the drawing content and aligned with the target subtask definition. We then apply multi-stage quality control to ensure that each question (i) satisfies hard constraints on format, scope, and subtask alignment, (ii) is faithful to drawing facts, and (iii) admits a unique and verifiable answer. After question validation, we answer each remaining question with multiple strong models and perform semantic majority voting; question-answer pairs without a clear majority agreement are discarded. The retained pairs are further assigned difficulty labels based on the complexity of both the drawing and the question. Finally, we conduct an expert audit on a stratified subset of the retained samples to verify question grounding, label correctness, and answer validity, and use the findings to refine prompts and filtering rules. In this sense, MechVQA prioritizes answerability and verifiability over brute-force scale: weakly grounded or ambiguous questions are revised or removed during validation.

**Source I: VQA free generation.** For open-ended coverage over diverse drawings, we adopt a free-form generation route using strong closed-source MLLMs (OpenAI, 2025; Google DeepMind, 2025a; Anthropic, 2025b). Given a drawing and its metadata, we randomly select one generator model and provide the subtask definitions. The generator first assesses the drawing complexity using both visual cues and metadata, and then produces a structured list of candidate questions together with subtask labels. To improve controllability and correctness, we perform iterative cross-model checking. In each round, we use a different model as a validator to check every question for grounding, unique answerability, and subtask consistency. The validator either accepts the question, rewrites it to fix violations, or rejects it as unanswerable. After validation, we answer each question with multiple strong models and retain a QA pair only if the candidate answers reach a clear semantic majority. We implement majority voting with a strong LLM judge (DeepSeek-AI, 2025), which compares the core semantics of candidate answers to determine the final decision.

**Source II: Template-based generation without ground truth.** To increase coverage for specific subtasks while maintaining diversity, we design subtask-specific templates

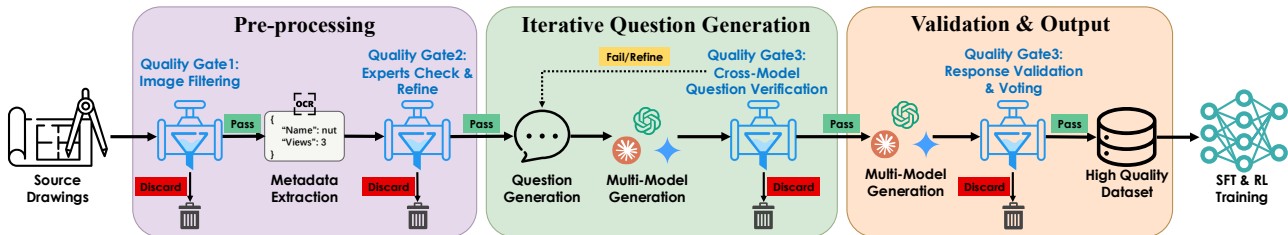

*(a)* Data construction pipeline of MechVQA.

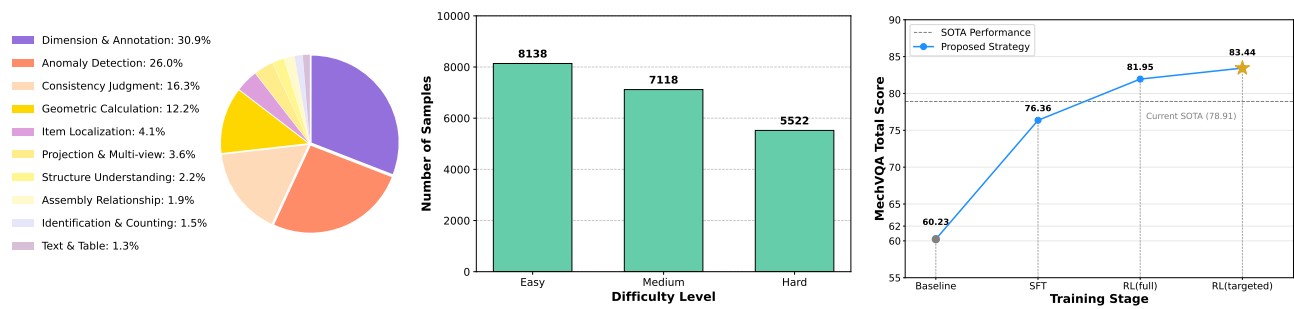

*(b)* Dataset statistics of MechVQA.

*(c)* Efficacy of Multi-Stage Training.

*Figure 2.* **MechVQA dataset construction and analysis.** (a) Source drawings and mechanical textbooks are filtered and annotated to obtain reliable metadata. Multiple strong MLLMs generate and self-refine QA pairs, which are further screened by voting-based quality checks and manual review, yielding the final MechVQA dataset used for evaluation and post-training. (b) MechVQA contains 10 subtasks with varying quantities and three difficulty levels ranging from easy to hard. (c) Efficacy of multi-stage training on MechVQA total score, comparing the base model, SFT, RL on full data, and targeted RL with self-play resampling.

that are instantiated from the drawing content. For example, for *Dimension & Annotation*, the model is prompted to first locate multiple dimension callouts and annotation symbols and then generate template questions by binding the symbol type and the referenced feature or region. Since these questions do not come with intrinsic ground-truth answers, we apply the same quality-control procedure as in VQA free generation, including multi-round cross-model question checking and multi-model answering with majority voting.

**Source III: Template-based generation with ground truth.** We further construct questions with answers directly supported by verified metadata or controlled expert edits. First, we generate metadata-grounded questions using deterministic templates, such as querying the number of views, the presence of section views, or other metadata fields. Because metadata are verified by experts, these templates provide reliable ground-truth answers. Second, we introduce human-expert-crafted problems. This includes 2D and 3D view matching questions constructed by pairing a 2D drawing with its correct isometric view or deliberately mismatched candidates. We also include anomaly and compliance questions created by experts using CAD tools to edit drawings, for example by adding redundant annotations, removing necessary dimensions, or injecting incorrect sizes and drafting symbols. These edits are limited to judging-oriented questions with known inconsistencies,

not full-corpus synthesis.

All answers are generated with a detailed explanation plus a final short answer format to support subsequent post-training, especially RL, and to improve the model's step-by-step analysis ability in mechanical drawing understanding. Even for questions with known ground-truth answers, we still prompt models to produce explicit reasoning to maintain consistency. Additional generation details and representative examples are provided in Appendix B.4, including the prompts for question generation, question checking, and answer voting.

### 3.4. Dataset Statistics

Across the three sources described above, we obtain 20,778 question-answer pairs, forming the MechVQA dataset. Figure 2b reports the distribution over subtasks and difficulty levels. Furthermore, we split MechVQA into train, validation, and test sets with an 8:1:1 ratio under strict drawing-level separation, where all QA records derived from the same drawing image are assigned to a single split to avoid leakage. To reduce near-duplicate overlap while keeping the label distributions similar, we compute a fused CLIP (Radford et al., 2021) representation per drawing group by combining mean-pooled text embeddings and an image embedding, cluster groups in the fused space, and then allocate clusters to train, validation, and test with stratifica-

tion over data source, subtask, and difficulty. This split protocol is designed to reduce benchmark-internal leakage from duplicated drawings and near-duplicate variants. Since MechVQA is built from public sources, absolute contamination cannot be ruled out, but we explicitly mitigate overlap at the drawing and similarity-cluster levels. The split results are presented in Appendix B.6.

# 4. MechVL: A Domain-Specialized Baseline

## 4.1. Supervised Fine-Tuning Stage

We initialize MechVL from Qwen3-VL-Instruct-4B (Yang et al., 2025) and perform full-parameter SFT, in contrast to PEFT (Houlsby et al., 2019) methods such as LoRA (Hu et al., 2022), exclusively on the LLM module, while keeping both the vision encoder and vision projection layers frozen. SFT is conducted on the MechVQA training split. Each training instance is a tuple $(x, q, y^*)$, where $x \in \mathcal{X} \cup \{\emptyset\}$ is an optional drawing image, $q$ is a drawing-grounded question, and $y^*$ is the target response following our unified schema with a rationale and a concise final answer. This exposure encourages schema compliance while grounding intermediate reasoning in the drawing content.

**Training objective.** We adopt the standard causal language modeling objective. Given $(x, q, y^*)$, the loss is

$$\mathcal{L}_{\text{SFT}} = -\sum_{t=1}^{T} \log \pi_\theta(y_t^* \mid x, q, y_{<t}^*), \tag{1}$$

where $T$ is the target length and $\pi_\theta$ denotes the model distribution. Optimizing the standard cross-entropy objective on MechVQA trains the model to ground drafting cues to mechanical semantics while following the required response schema. The resulting reference policy $\pi_{\text{ref}}$ produces convention-consistent, professionally phrased answers and serves as the initialization for the subsequent RL stage.

## 4.2. Reinforcement Learning Stage

**DAPO.** We further optimize MechVL via RL using *Decoupled Clip and Dynamic Sampling Policy Optimization* (DAPO) (Yu et al., 2025). DAPO builds on group-based policy optimization (e.g., GRPO) and introduces four techniques to improve training stability and efficiency for long-form reasoning.

Let $u = (x, q)$ denote a multimodal prompt consisting of a drawing image $x$ and a question $q$, and assume $u$ is associated with a verifiable answer $a$. At each update, we sample a group of $G$ candidate responses $\{y_i\}_{i=1}^{G}$ from an old policy $\pi_{\theta_{\text{old}}}(\cdot \mid u)$. Each response $y_i$ receives a scalar reward $R_i = r(u, y_i)$. Following DAPO, we compute a group relative advantage that is shared across all tokens of

$y_i$:

$$\hat{A}_{i,t} = \frac{R_i - \text{mean}\left(\{R_j\}_{j=1}^{G}\right)}{\text{std}\left(\{R_j\}_{j=1}^{G}\right) + \epsilon_A}, \tag{2}$$

where $\epsilon_A$ is a small constant for numerical stability and the subscript $t$ emphasizes that the same group normalized advantage is used for every token of $y_i$.

DAPO uses a token-level policy gradient surrogate. Specifically, it defines the importance ratio at the *token-level*:

$$r_{i,t}(\theta) = \frac{\pi_\theta(y_{i,t} \mid u, y_{i,<t})}{\pi_{\theta_{\text{old}}}(y_{i,t} \mid u, y_{i,<t})}, \tag{3}$$

and optimizes a PPO-style clipped surrogate aggregated over all tokens in the group:

$$\mathcal{L}_{\text{DAPO}} = -\mathbb{E}_{u, \{y_i\}} \left[ \frac{1}{\sum_{i=1}^{G} |y_i|} \sum_{i=1}^{G} \sum_{t=1}^{|y_i|} \min \left( r_{i,t}(\theta) \, \hat{A}_{i,t}, \right. \right.$$
$$\left. \left. \text{clip}(r_{i,t}(\theta), 1 - \epsilon_{\text{low}}, 1 + \epsilon_{\text{high}}) \, \hat{A}_{i,t} \right) \right], \tag{4}$$

where DAPO applies *decoupled* clipping (Clip Higher) with separate lower and upper ranges, controlled by $\epsilon_{\text{low}}$ and $\epsilon_{\text{high}}$.

To avoid degenerate groups that provide little learning signal, DAPO adopts *Dynamic Sampling*. Concretely, each prompt retains both positively and negatively rewarded samples:

$$0 < \left| \{y_i \mid \text{is\_equivalent}(a, y_i)\} \right| < G, \tag{5}$$

implemented by repeatedly sampling and discarding groups where all $G$ responses are either correct or incorrect.

Finally, DAPO uses *Overlong Reward Shaping* when computing $r(u, y)$ to handle responses that exceed the length budget, reducing reward noise introduced by truncation or overly long generations. Following the standard DAPO setup, we omit an explicit KL penalty to the SFT reference policy and rely on the rule-based reward, asymmetric clipping, Dynamic Sampling, and overlong shaping to regularize training for long-form multimodal generation.

**Reward design.** For mechanical drawing VQA, we design a composite reward that balances verifiable correctness, strict schema compliance, and explanation quality:

$$R(x, q, y) = \lambda_{\text{acc}} \, r_{\text{acc}}(x, q, y)$$
$$+ \lambda_{\text{fmt}} \, r_{\text{fmt}}(y) + \lambda_{\text{qual}} \, r_{\text{qual}}(x, q, y), \tag{6}$$

where $r_{\text{acc}} \in [0, 1]$, $r_{\text{fmt}} \in \{0, 1\}$, and $r_{\text{qual}} \in [0, 1]$ and $\lambda_{\text{acc}}, \lambda_{\text{fmt}}, \lambda_{\text{qual}}$ are coefficients that balance each term.

- **Accuracy reward.** We extract the final answer from response and compare it with the ground-truth answer

$a^*$, and then obtain $r_{\text{acc}}$. This term quantifies technical correctness of model response.

- **Format reward.** To enforce strict output structure for downstream parsing and evaluation, we use a binary format reward commonly used in RLVR paradigm. The reward is 1 only if the response can be parsed as a well-formed output that contains exactly one rationale span enclosed by `<think>...</think>` and a answer span enclosed by `<answer>...</answer>`; otherwise it is 0. This prevents degenerate generations that omit either the rationale or the final answer and improves training stability.

- **Quality reward.** To improve overall response quality, we use LLM-as-a-Judge with an explicit rubric that scores three axes: *Logic* (coherence and self-consistency), *Professionalism* (correct mechanical-drawing terminology and technical phrasing), and *Conciseness* (avoiding redundancy and off-topic content):

$$r_{\text{qual}} = \frac{s_{\text{logic}} + s_{\text{prof}} + s_{\text{conc}}}{3}, \qquad (7)$$

where $s_{\text{logic}}, s_{\text{prof}}, s_{\text{conc}} \in [0, 1]$.

Remarkably, instead of strict string matching, we also employ LLM-as-a-Judge for Accuracy Reward to assess semantic equivalence and return a score in $[0, 1]$. This gives non-zero reward to answers that are semantically correct but expressed differently, while penalizing technically incorrect or mismatched answers.

**Two-stage self-play RL.** We run DAPO in a self-play style. First, we perform RL on the full MechVQA training split. Then we train the model on a sampled subset with an increased proportion of underperforming subtasks, while keeping the same objective in Eq. 4 and reward in Eq. 6. Figure 2c also shows the efficacy of self-play RL.

Overall, $r_{\text{acc}}$ anchors domain correctness with semantic-tolerant matching, $r_{\text{fmt}}$ guarantees schema validity for stable training and evaluation, and $r_{\text{qual}}$ promotes mechanically grounded, professional, and concise explanations while reducing spurious correctness. Together, they directly address common failure modes on dense mechanical drawings such as missing decisive annotations, cross-view inconsistencies, and answers that appear numerically plausible but violate drawing constraints.

## 5. Experiments

### 5.1. Experimental Setup

**Evaluation.** We evaluate MechVL and all baselines on the MechVQA test split constructed in Section 3.4 and use accuracy as the primary metric. To ensure scalable, reproducible, and robust evaluation, we employ multiple strong LLMs as automatic judges, including GPT-OSS-120B (OpenAI, 2025), DeepSeekV3.2 (Guo et al., 2025), and Kimi-k2 (Team et al., 2025b) for each test question. Details of the judge prompt, score aggregation, and failure handling are provided in Appendix C.4.

**Implementation Details.** We initialize MechVL from Qwen3-VL-4B-Instruct. For SFT, we use the LLaMA-Factory (Zheng et al., 2024) framework and perform full-parameter training. For RL, we adopt the EasyR1 (Zheng et al., 2025) framework. The full hyperparameter configuration is provided in the Appendix C.1 and C.2.

**Baselines.** To benchmark the effectiveness of MechVL, we compare against a broad set of strong general-purpose MLLMs, covering both open-source models and closed-source APIs. Our open-source baselines span multiple recent model families and scales, including Qwen3-VL models from 4B to 32B (Yang et al., 2025), as well as GLM-4.6V (Team et al., 2025c), InternVL3.5 (Wang et al., 2025), MiniCPM-V (Yao et al., 2024), MiMo-VL (Xiaomi, 2025), Llama-Vision (AI, 2024), and Gemma (Team et al., 2025a). Our closed-source baselines include GPT-4o, GPT-4o-mini (OpenAI, 2024) and GPT-5 (OpenAI, 2025), Gemini 3 Pro Preview (Google DeepMind, 2025b), Claude Sonnet 4.5 (Anthropic, 2025a), and Qwen3-VL-Plus (Bai et al., 2023). All baselines are evaluated under the same protocol on MechVQA without external tools, retrieval, or additional domain adaptation. For general-purpose baselines, this measures out-of-domain transfer; MechVL instead serves as an in-domain baseline quantifying targeted post-training.

### 5.2. Main Results

Table 2 reports the main results on MechVQA. **MechVL-4B-RL** achieves the best overall score of **84.85**, outperforming the strongest open-source baseline (GLM-4.6V, 78.91) by **+5.94** and closed-source baseline (Gemini-3-Pro-Preview, 77.28) by **+7.57**. Relative to **MechVL-4B-SFT** (76.36), RL brings a large gain of **+8.49**, indicating that self-play RL is essential for improving reliability on dense drawings and constraint-sensitive tasks. **MechVL-4B-RL** also attains the best scores on **DA** (90.70), **IL** (82.01), **SU** (83.33), **AR** (84.00), **PM** (64.00), and **AD** (86.94), suggesting that domain post-training preserves strong perception while substantially strengthening projection-consistent reasoning and standards-aware judgment.

**Difficulty stratification.** Figure 3 reports accuracy on the easy, medium, and hard subsets (in percentage). Since the three subsets contain different numbers of questions, rankings in this figure do not necessarily match the overall total score in Table 2; we use it mainly to compare how well a model stays balanced across difficulty levels. All models exhibit a clear accuracy drop from easy to hard, indicating that harder questions require more reliable cross-view cor-

| Model | Recognition | | | | Reasoning | | | | Judging | | Total |
|---|---|---|---|---|---|---|---|---|---|---|---|
| | IC | DA | TT | IL | SU | GC | AR | PM | AD | CJ | |
| **Open-source MLLMs** | | | | | | | | | | | |
| Qwen3-VL-4B-Instruct | 62.79 | 76.96 | 88.64 | 33.09 | 39.58 | 58.54 | 22.00 | 20.00 | 62.86 | 49.33 | 60.23 |
| InternVL3.5-8B | 65.12 | 61.10 | 77.27 | 19.42 | 37.50 | 48.78 | 28.00 | 34.00 | 53.47 | 34.67 | 49.82 |
| MiniCPM-V-4.5 | 79.07 | 76.32 | 84.09 | 30.22 | 45.83 | 59.76 | 24.00 | 36.00 | 60.00 | 46.67 | 60.51 |
| Mimo-VL-7B-SFT | 79.07 | 76.74 | 93.18 | 38.85 | 27.08 | 54.47 | 26.00 | 20.00 | 59.59 | 46.67 | 59.66 |
| Llama-3.2-11B-Vision-Instruct | 13.95 | 38.05 | 54.55 | 11.51 | 6.25 | 25.61 | 0.00 | 8.00 | 17.55 | 28.00 | 25.48 |
| Qwen3-VL-30B-A3B-Instruct | 74.42 | 78.01 | 90.91 | 37.41 | 43.75 | 64.23 | 38.00 | 40.00 | 64.08 | 57.33 | 64.47 |
| Gemma-3-27B-it | 46.51 | 55.39 | 70.45 | 24.46 | 31.25 | 51.22 | 24.00 | 24.00 | 24.90 | 40.00 | 42.68 |
| Qwen3-VL-32B-Instruct | 86.05 | 86.68 | **97.73** | 56.83 | 62.50 | 76.83 | 62.00 | 48.00 | 75.92 | 69.33 | 76.50 |
| GLM-4.6V | **88.37** | 86.68 | **97.73** | 63.31 | 77.08 | **82.93** | 60.00 | 62.00 | 74.29 | 69.33 | 78.91 |
| **Closed-source MLLMs** | | | | | | | | | | | |
| GPT-4o | 62.79 | 75.90 | 81.82 | 40.29 | 64.58 | 63.41 | 48.00 | 44.00 | 53.06 | 66.67 | 63.06 |
| GPT-4o-mini | 32.56 | 61.52 | 68.18 | 23.02 | 29.17 | 47.97 | 24.00 | 22.00 | 35.10 | 49.33 | 45.65 |
| Qwen3-VL-Plus | 81.40 | 86.68 | 95.45 | 52.88 | 81.25 | 79.67 | 74.00 | 58.00 | 67.76 | 68.00 | 76.33 |
| GPT-5 | 69.77 | 84.99 | 93.18 | 62.59 | 79.17 | 73.58 | 60.00 | 60.00 | 71.02 | 70.67 | 75.44 |
| Gemini-3-Pro-Preview | 76.74 | 87.74 | **97.73** | 64.03 | 52.08 | 73.58 | 46.00 | 58.00 | 78.37 | **82.67** | 77.28 |
| Claude-Sonnet-4.5 | 67.44 | 78.65 | **97.73** | 56.12 | 70.83 | 75.20 | 62.00 | 54.00 | 64.90 | 64.00 | 71.20 |
| **MechVL** | | | | | | | | | | | |
| MechVL-4B-SFT (Ours) | **88.37** | 85.20 | **97.73** | 61.15 | 60.42 | 73.17 | 40.00 | 44.00 | 84.49 | 69.33 | 76.36 |
| **MechVL-4B-RL (Ours)** | **88.37** | 90.70 | **97.73** | 82.01 | 83.33 | 76.83 | **84.00** | 64.00 | 86.94 | 78.67 | **84.85** |

*Table 2.* Evaluation results of MechVQA on open-source, closed-source MLLMs and our MechVL models. **Bold** denotes the best scores on subtasks or total. The MechVL model with the best overall performance is highlighted in ⬚ purple ⬚.

respondence, stricter constraint satisfaction, and multi-step reasoning beyond direct reading. **MechVL-4B-RL** achieves the strongest and most balanced performance across all three levels, reaching **94%** (easy), **79%** (medium), and **75%** (hard). Compared with **MechVL-4B-SFT**, RL yields the largest gains on the harder subsets, improving medium accuracy from 70% to 79% and hard accuracy from 53% to 75%, while keeping easy performance similar (92% to 94%). It also outperforms the strongest closed-source baseline on the hard subset (Qwen3-VL-Plus, 66%) by 9 percentage points. Overall, the gains concentrate on medium and hard questions, showing that RL mainly improves robustness under higher reasoning and consistency demands.

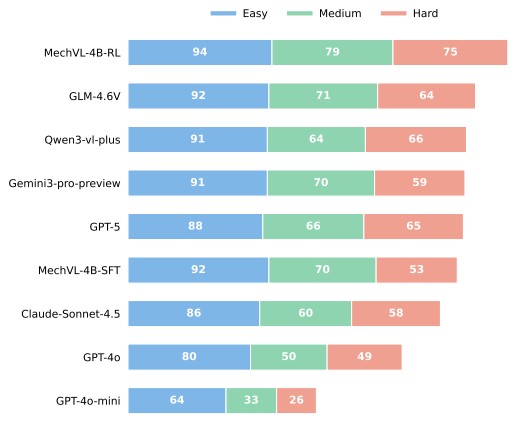

*Figure 3.* Evaluation results on difficulty levels

**Capability-wise comparison.** Beyond the overall score, MechVL-4B-RL shows consistent gains across all three capability axes. Averaging the corresponding subtasks in Table 2, MechVL-4B-RL reaches 89.70 on *Recognition*, 77.04 on *Reasoning*, and 82.81 on *Judging*. Compared with GLM-4.6V, this corresponds to gains of +5.68, +6.54, and +11.00, respectively; compared with Gemini-3-Pro-Preview, the gains are +8.14, +19.62, and +2.29. These results show that post-training consistently strengthens perception-heavy reading, projection-consistent reasoning, and standards-aware decision making.

### 5.3. Ablation Study

We conduct two ablations to isolate the effect of (i) multi-stage RL post-training and (ii) the RL optimizer choice. Following the MechVQA capability taxonomy, we report capability-wise mean scores, together with the total score and the mean over all ten subtasks.

**SFT vs. SFT+RL.** Table 3(A) compares the SFT-only model with two RL variants. Moving from SFT to full-data DAPO yields a substantial improvement, increasing the *Total* score from 76.36 to 81.95 and the *Avg.* score from 70.39 to 79.12. The gain is primarily driven by *Reasoning*, which rises markedly from 54.40 to 70.75, while *Recognition* also improves from 83.11 to 86.26 and *Judging* increases from 76.91 to 81.62. Further applying targeted RL, which up-weights underperforming subtasks during sampling, brings additional gains across all three capability axes, achieving the best overall performance with a *Total* score of 84.85

| Setting | Rec. | Reas. | Judg. | Total | Avg. |
|---|---|---|---|---|---|
| **(A) Training stages** | | | | | |
| SFT | 83.11 | 54.40 | 76.91 | 76.36 | 70.39 |
| + DAPO (full) | 86.26 | 70.75 | 81.62 | 81.95 | 79.12 |
| + DAPO (targeted) | **89.70** | **77.04** | **82.81** | **84.85** | **83.26** |
| **(B) RL algorithms in full phase** | | | | | |
| GRPO | 83.55 | 64.49 | 77.93 | 80.47 | 74.80 |
| GSPO | 84.17 | 61.29 | 77.73 | 78.77 | 73.73 |
| DAPO | **86.26** | **70.75** | **81.62** | **81.95** | **79.12** |
| **(C) Reward design in targeted phase** | | | | | |
| Acc(0/1) | 86.62 | 71.72 | 79.42 | 82.24 | 79.22 |
| Acc(F1) | 85.49 | 65.88 | 79.32 | 80.33 | 76.41 |
| w/o Qual | 88.38 | **77.23** | 81.68 | 83.44 | 82.58 |
| Full | **89.70** | 77.04 | **82.81** | **84.85** | **83.26** |

*Table 3.* Combined ablation results on training stages, RL algorithms, and reward design. Rec./Reas./Judg. denote capability-wise mean scores over subtasks; Avg. is the mean over all ten subtasks.

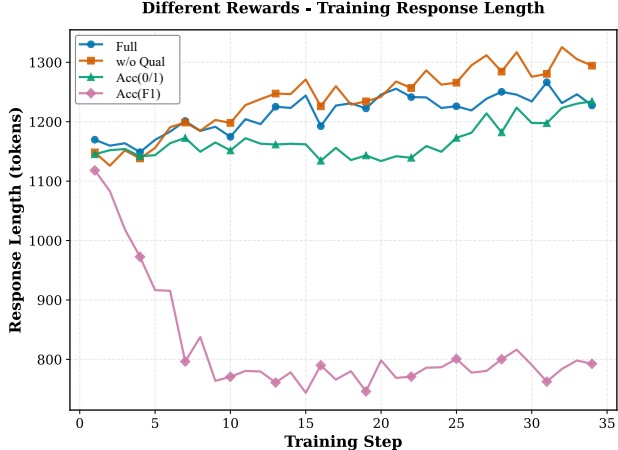

*Figure 4.* Response length dynamics under different reward designs during RL training. Acc(F1) rapidly shortens responses, indicating that token-overlap feedback encourages terse outputs; w/o Qual produces the longest traces, reflecting verbosity without a matching accuracy gain. The full reward maintains controlled response lengths while achieving the best final performance, suggesting better-calibrated reasoning traces.

and an *Avg.* score of 83.26. Notably, targeted RL improves *Reasoning* from 70.75 to 77.04 and *Recognition* from 86.26 to 89.70, suggesting that focusing updates on weaker categories helps reduce capability imbalance beyond full-data RL.

**RL algorithm ablation.** Table 3(B) compares GRPO, GSPO, and DAPO under the same initialization (MechVL-4B-Inst) and training setting. DAPO achieves the best overall results, with a *Total* score of 81.95 and an *Avg.* score of 79.12, outperforming GRPO (80.47 / 74.80) and GSPO (78.77 / 73.73). DAPO leads all three capability-wise averages, with the largest margin on *Reasoning*, where it reaches 70.75 compared with 64.49 for GRPO and 61.29 for GSPO; it also delivers consistent gains on *Recognition* and *Judging*. These results indicate that DAPO provides a more effective optimization signal for mechanically grounded reasoning and constraint-sensitive behaviors in dense drawings.

**Reward design ablation.** We ablate the reward design in Stage 2 targeted self-play RL while keeping the training setting fixed. Table 3(C) compares Acc(0/1), Acc(F1), w/o Qual, and Full (Eq. 6). Full performs best, reaching 84.85 on Total and 83.26 on Avg.; removing quality drops Total to 83.44, binary accuracy reaches 82.24, and token-level F1 performs worst at 80.33. These results show that the reward should combine semantic correctness, schema compliance, and explanation quality rather than relying on coarse or surface-level matching. Appendix C.3 further reports reward-weight ablations.

**Reward dynamics.** Figure 4 visualizes response-length dynamics under different reward designs. Acc(F1) quickly collapses from roughly 1.1K tokens to below 0.8K, indi-

cating that overlap-based feedback favors terse but weakly grounded responses. In contrast, w/o Qual drifts toward the longest outputs around 1.3K tokens, reflecting verbosity without commensurate correctness gains. Acc(0/1) is less stable than the full reward, while Full maintains a controlled band around 1.2K–1.25K tokens and achieves the best final accuracy in Table 3(C), suggesting better-calibrated reasoning traces rather than merely longer outputs.

## 6. Conclusion

In this paper, we studied mechanical drawing understanding with MLLMs under orthodox drafting conventions, where correct interpretation requires dense visual reading, projection-consistent spatial understanding, multi-step geometric reasoning, and standards-aware judgment. We introduced **MechVQA**, a benchmark of real part and assembly drawings with ten subtasks across *Recognition*, *Reasoning*, and *Judging*. We further established **MechVL**, a domain-specialized baseline trained with supervised fine-tuning followed by DAPO-based reinforcement learning. Experiments show consistent gains over strong MLLMs, especially on reasoning-intensive and standards-sensitive subtasks.

## Impact Statement

This project advances AI understanding and reasoning in the domain of engineering drawings, a field traditionally reliant on specialized human expertise. By introducing MechVQA—a dedicated, fine-grained benchmark—and the

MechVL model enhanced via SFT and DAPO-based reinforcement learning, our work enables the systematic evaluation and development of models capable of interpreting complex mechanical diagrams.

The potential benefits of this research are multi-fold:

- **Industrial Productivity**: Automating the interpretation of standardized symbols and multi-view projections can significantly streamline design reviews and inspection workflows.

- **Error Reduction**: By assisting engineers in cross-checking complex dimension chains and geometric tolerances, MechVL helps mitigate human oversights in high-density technical documents.

- **Knowledge Accessibility**: Our work lowers the barrier for non-experts and students to comprehend structured engineering language, supporting interdisciplinary collaboration and technical learning accessibility in manufacturing.

Nonetheless, we recognize that over-reliance on automated interpretations carries inherent risks. Given that mechanical design often dictates structural integrity, any misuse or blind trust in model outputs could lead to design flaws or even safety-critical failures. To mitigate these risks, we emphasize that MechVL is intended as a decision-support assistant rather than a replacement for certified professional judgment. We strongly recommend maintaining rigorous human oversight and verification within any integrated engineering workflow.

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

# A. Limitations

Despite the advancements presented by MechVQA and MechVL, several limitations remain.

- **Scope of source drawings.** MechVQA is built from publicly available educational and professional materials, including textbooks, handbooks, and design platforms, rather than proprietary industrial archives. As a result, it may not fully capture the variability of real factory drawings, legacy blueprints, or company-specific drafting conventions.

- **Focus on 2D drawing understanding.** Although MechVQA includes multi-view reasoning and questions involving isometric views, the benchmark is centered on drawing-grounded understanding of 2D mechanical drawings. It does not aim to solve full 3D CAD reconstruction or direct generation of engineering file formats such as STEP or IGES.

- **Dependence on OCR and visual clarity.** The benchmark construction pipeline relies partly on OCR, metadata extraction, and expert verification. While we apply multi-stage validation, semantic voting, and expert audit, performance may still degrade on drawings with extreme annotation clutter, poor scan quality, or visually ambiguous local regions.

- **Residual contamination risk.** We explicitly mitigate benchmark-internal leakage by enforcing drawing-level split and similarity-aware allocation. However, because the benchmark is constructed from public sources and modern foundation models are trained on broad web-scale corpora, absolute contamination cannot be ruled out.

- **Benchmark validation is still incomplete.** We currently do not report human-expert upper bounds or formal inter-annotator agreement statistics. However, the annotation process is guided by a written annotation handbook and structured workflow, rather than ad hoc labeling. Future releases will include human agreement analysis to better quantify benchmark difficulty, annotation consistency, and the remaining gap between models and domain experts.

- **Release and licensing.** Since MechVQA is constructed from publicly accessible educational and professional sources, the release will follow the redistribution permissions of the underlying materials. When direct redistribution of original drawings is restricted, we will release the corresponding annotations, metadata, split information, prompts, and source references whenever permitted.

# B. MechVQA Details

## B.1. Example of Mechanical Drawings and Metadata

Figure 5 shows an example mechanical drawing. The corresponding metadata schema is summarized in Table 4, and a concrete metadata example is provided in Figure 6.

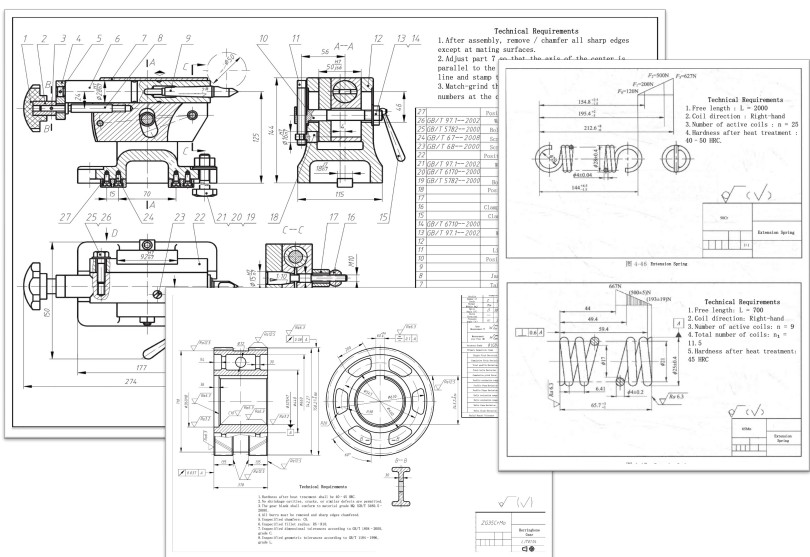

*Figure 5.* Example of mechanical drawings from MechVQA Dataset.

| Metadata | Scope | Description |
|---|---|---|
| Image type | Global | Single or multiple drawings in one image. |
| Name | Part, Assembly | Name of drawing. |
| Drawing type | Part, Assembly | Part or assembly. |
| View count | Part, Assembly | Number of views. |
| Primary views | Part, Assembly | Number of front, side and top views. |
| Special views | Part, Assembly | Number of isometric, enlarged, section, exploded, local and directional views. |
| Textual content | Part, Assembly | Technical requirements, title block, parameter table and BOM(Bill of Materials). |
| Part category | Part | Category label of part drawing. |
| Sub-parts | Assembly | Sub-parts of an assembly include the above corresponding attributes. |

*Table 4.* Metadata schema for MechVQA.

---

**Example of Metadata Extracted from Mechanical Drawings**

**Image Path:** xxx.png

**Drawing Parameters:**

- **Composition Type:** Single Sheet

- **Drawing Category:** Part Drawing

- **Part Name:** Center Mandrel

- **Standard Part:** No

- **Part Type:** Others

- **Layout Type:** Multi-view

- **Number of Views:** 3

**Main Views:**

- Front View: 1

- Side View: 1

- Top View: 1

**Special Views:**

- Section View: 1

- Axonometric View: 0

- Enlarged View: 0

- Exploded View: 0

- Local View: 0

- Auxiliary View: 0

- Broken View: 0

**Technical Requirements:**

1. The tip head is carburized and quenched, hardness 40–45 HRC.

2. Unspecified chamfer: C1.

3. Unspecified dimension tolerance according to GB/T1804–2000–m.

4. Unspecified geometric tolerance according to GB/T1184–1996–K.

---

*Figure 6.* Example of metadata extracted from a mechanical drawing

## B.2. Effect of Expert Verification

To quantify the role of expert verification, we compare model-extracted metadata before human checking with the final expert-corrected metadata on the audited typical and newstandard groups. We exclude bookkeeping-style schema migration fields from this analysis and focus on mechanically meaningful corrections. As summarized in Table 5, expert checking substantially affects view counting, view-type classification, and technical-requirement transcription. The dominant errors are not random: side and top views are systematically over-labeled, directional and enlarged views are often missed, and section views are frequently confused with adjacent special-view categories. Technical requirements also require substantial

manual correction, often due to inaccurate surface-treatment descriptions, heat-treatment parameters, defect-type wording, and chamfer notation. In contrast, part-category labels are comparatively stable, changing in less than 1% of audited cases. These findings confirm that expert verification is not merely cosmetic, but improves the reliability of structured metadata used for downstream question construction.

| Field group | Correction rate | Main pattern |
| --- | --- | --- |
| View count | 41.6% | The model usually undercounts views; expert correction often increases the count. |
| Side / top views | 33.0% / 31.8% | The model over-labels side and top views, mistaking local or auxiliary regions for major views. |
| Section views | 37.8% | The model confuses section views with section-like or directional views. |
| Directional / enlarged views | 25.4% / 11.5% | The model mostly misses directional and enlarged views rather than hallucinating them. |
| Technical requirements | 43.7% | The model misstates surface treatments, heat-treatment parameters, defect descriptions, and chamfer notation. |
| Part category | 1.0% | The model is mostly reliable on coarse part-category labels. |

*Table 5.* Effect of expert verification on metadata extraction. Only correction rates are reported.

## B.3. Task Taxonomy and Difficulty Level Definitions

Detailed subtask definition is shown in 7 and the definition of difficulty levels is provided in 8.

---

**Task Taxonomy Definition for MechVQA**

**Recognition.** The ability to accurately extract explicit visual information from engineering drawings, including textual, symbolic, dimensional, and spatial content.

- **IC (Identification & Counting):** Identify specified entities and count instances such as parts, features, symbols, or views (e.g., counting holes, identifying part names).

- **DA (Dimension & Annotation):** Read dimensions and drafting annotations (e.g., datums, surface roughness, tolerances, and callouts) and ground them to the referenced geometric features (e.g., tolerance reading, datum identification).

- **TT (Text & Table):** Interpret textual blocks and tabular content such as title blocks, parameter tables, technical requirements, and BOM entries (e.g., title-block field extraction, BOM lookup).

- **IL (Item Localization):** Localize target views or regions and ground the referenced object, feature, or annotation to its spatial position within the drawing (e.g., view or region localization).

**Reasoning.** The ability to infer implicit relationships and unannotated information based on explicit geometric structure, dimensional constraints, and multi-view correspondences.

- **SU (Structure Understanding):** Infer drawing structure and semantics, including feature composition and relations induced by sectional views and view organization (e.g., section-structure reasoning).

- **GC (Geometric Calculation):** Compute target quantities through dimensional chains and geometric constraints rather than direct reading (e.g., dimension-chain calculation).

- **AR (Assembly Relationship):** Infer assembly relations such as mating conditions, relative placement, functional interactions, and part dependencies (e.g., interference or fit reasoning).

- **PM (Projection & Multi-view):** Reason across multiple views via cross-view correspondence under projection-consistent constraints (e.g., multi-view correspondence analysis).

**Judging.** The ability to assess the correctness, consistency, and standards compliance of engineering drawings.

- **AD (Anomaly Detection):** Detect missing, conflicting, or implausible dimensions, annotations, view relations, or table entries (e.g., missing dimensions, contradictory annotations).

- **CJ (Consistency Judgment):** Judge global consistency and compliance with drafting standards and design constraints (e.g., standards or specification compliance checks).

*Figure 7.* Task taxonomy of MechVQA

---

**Difficulty Level Definition for MechVQA**

**Simple.** Basic recognition and direct information extraction without reasoning.

- **Characteristics:** Single explicit information reading; clear dimensions, symbols, or text; simple naming, localization, or counting; no computation or domain knowledge.

- **Answer Style:** Concise factual answers in 1–2 sentences.

- **Examples:** Reading labeled dimensions, identifying part names or symbols, counting holes, surface roughness lookup.

**Medium.** Requires understanding and moderate reasoning across multiple information sources.

- **Characteristics:** Integration of 2–3 cues; cross-view correspondence; simple geometric or dimensional inference; basic structural or assembly understanding.

- **Answer Style:** Short analytical explanations (3–5 sentences) with clear inference basis.

- **Examples:** Computing unannotated dimensions, inferring internal structures from section views, analyzing basic fit or assembly relations.

**Hard.** Requires deep analysis and comprehensive reasoning supported by professional knowledge.

- **Characteristics:** Multi-view and distributed information integration; complex spatial reasoning; multi-step dimension-chain analysis; consideration of materials, standards, and design constraints.

- **Answer Style:** Complete, well-structured analysis with domain justification (5+ sentences).

- **Examples:** Tolerance-chain computation, derivation of missing dimensions via dimensional analysis, identification of design weaknesses.

*Figure 8.* Difficulty Level definition for MechVQA

## B.4. Generation Prompts for MechVQA

Please carefully analyze the given mechanical drawing image and generate high-quality, professional queries strictly based on the visual content of the drawing.
**Image Analysis Instructions**

- First, analyze the structure of the image, including the number of views or sub-figures.

- Determine whether the image contains:a single drawing, multiple related views of the same part, or multiple independent drawings.

- If multiple independent sub-figures exist, each query must explicitly specify the target sub-figure (e.g., "top-left drawing", "second sub-figure").

- Ignore surrounding textual content such as page numbers, captions, or problem statements.

**Query Capability Types** Each query must belong to exactly one of the following capability dimensions:

- **Recognition**: Direct extraction of explicit information from the drawing (e.g., dimensions, symbols, labels, counts).

- **Reasoning**: Queries requiring multi-view understanding, geometric inference, dimensional calculation, or structural interpretation.

- **Judgment**: Queries involving engineering judgment, such as design rationality, standard compliance, or potential structural issues.

**Difficulty Levels** Each query must be assigned one difficulty level:

- **Easy**: Direct reading or simple identification without inference.

- **Medium**: Requires integrating multiple visual cues or basic mechanical drawing knowledge.

- **Hard**: Requires complex reasoning, multi-step inference, or professional mechanical engineering knowledge.

The difficulty level should be consistent with both the drawing complexity and the cognitive demand of the query.
**Query Generation Requirements**

- Generate queries only if the required information is visually supported by the drawing.

- Do not hallucinate missing dimensions, structures, or annotations.

- Queries must be clear, precise, and professionally phrased.

- Queries should have practical engineering meaning.

**Output Format** Return the result strictly in the following JSON format, without any additional text:

```
{
  "Drawing Complexity": "Simple / Medium / Complex",
  "Number of Queries": N,
  "Queries": [
    ["Query text", "Capability", "Subcategory", "Difficulty", "Language"],
    ...
  ]
}
```

*Figure 9.* Question generation prompt for MechVQA without ground truth

**Role:** You are a professional mechanical engineer with strong expertise in interpreting mechanical drawings.

**Task:** Carefully analyze the given mechanical drawing and generate high-quality, professional questions related to *dimension and annotation recognition*. All questions must be strictly grounded in the actual visual content of the drawing.

**Important Instructions:**

1. Generate questions **only based on what is explicitly shown in the drawing**. Do not hallucinate or infer features that are not present.

2. **Ignore all surrounding textual descriptions** (e.g., title blocks, notes, tables). Focus exclusively on the graphical content of the drawing.

3. If the drawing contains **multiple sub-figures**, you must clearly specify which sub-figure each question refers to.

4. **Prioritize question quality over quantity**. Do not generate vague, trivial, or unanswerable questions.

**Given Drawing Metadata (Provided as Annotations):**

- **Combination Type:** Single drawing or Multiple sub-figures

- **Drawing Type:** {drawing_type} (provided metadata, used as reference)

- **View Composition:** {view_info} (provided metadata, used as reference)

**Note:** The above metadata is *explicitly given* and should not be inferred from the image. Use it only to assist interpretation, not as a source of new information.

**Current Objective:** Identify locations in the drawing that include explicit dimension annotations and use them to construct precise, professional questions.

**Procedure:**

1. Inspect the drawing and identify up to **five distinct locations or features** that are explicitly annotated with dimensions.

2. For each identified location, determine:

    - The **dimension value** (e.g., $\Phi30$, 50, $R5$, $15°$).
    - The **annotated location or feature** (e.g., "right end of the shaft", "bottom hole", "top fillet").
    - The **view** in which the dimension appears (e.g., front view, top view, side view, section A–A, detail view B).

3. Prefer diversity in both **locations** and **dimension types** (e.g., diameter, length, radius, chamfer, angle).

4. If fewer than five dimension annotations are present, report as many as can be reliably identified.

**Output Format:** Return the result strictly in the following JSON format, without any additional explanation or commentary:

```
{
  "dimensions": [
    {
      "dimension_value": "e.g., Phi30",
      "location": "annotated feature or position",
      "view": "corresponding view name"
    }
  ]
}
```

*Figure 10.* Question generation prompt for MechVQA with ground truth: Dimension

**Role:** You are a professional mechanical engineer with strong expertise in interpreting engineering drawings and manufacturing annotations.

**Task:** Carefully analyze the given mechanical drawing and generate high-quality, professional questions related to *engineering annotation recognition*. All questions must be strictly grounded in the actual visual content of the drawing.

**Important Instructions:**

1. Generate questions **only based on annotations that explicitly appear in the drawing**. Do not hallucinate missing symbols or values.

2. **Ignore all surrounding textual descriptions** (e.g., title blocks, technical notes, tables). Focus exclusively on graphical annotations within the drawing views.

3. **Prioritize correctness and clarity**. Do not generate vague, redundant, or unanswerable questions.

**Given Drawing Metadata (Provided as Annotations):**

- **Combination Type:** Single drawing or Multiple sub-figures

- **Drawing Type:** {drawing_type} (provided metadata, used as reference)

- **View Composition:** {view_info} (provided metadata, used as reference)

**Current Objective:** Identify and reason about different types of engineering annotations appearing in the drawing, and generate precise, professional questions for each identified annotation.

**Annotation Types to Consider:** Identify the following types of annotations whenever they appear in the drawing:

- **Datum symbols** (e.g., boxed capital letters such as A, B, C)

- **Datum features/locations** (points, lines, or planes associated with datum symbols)

- **Geometric tolerances** (GD&T symbols, tolerance values, and their meanings relative to datums)

- **Limit dimensions** (e.g., $50 \pm 0.2$, $\Phi 30^{+0.02}_{-0.01}$)

**Procedure:**

1. Inspect the drawing and identify existing annotations from the categories above.

2. For each identified annotation, generate one clear and professional question.

3. Each question must explicitly include:

   - The **annotation type** (datum symbol, datum feature, GD&T, limit dimension, roughness, etc.); The **annotation content** (specific symbol, value, or notation). The **location** (which view and which feature of the part).

4. If a specific annotation type does **not** appear in the drawing, do not generate a question for that type.

**Output Format:** Return the result strictly in the following JSON format, without any additional explanation or commentary:

```
{
  "annotations": [
    {
      "type": "annotation type (symbol/feature/roughness/other)",
      "content": "specific symbol or value",
      "location": "corresponding view and feature",
      "question": "generated professional question"
    }
  ]
}
```

*Figure 11.* Question generation prompt for MechVQA with ground truth: Annotation

**Role:** You are a professional mechanical engineer with extensive experience in reading and interpreting mechanical and assembly drawings.

**Task:** Carefully analyze the given mechanical drawing and generate high-quality, professional questions related to *view identification and spatial/location recognition*. All questions must be strictly grounded in the actual visual content of the drawing.

**Key Instructions:**

1. Generate questions **only based on elements that explicitly appear in the drawing**. Do not infer nonexistent views, annotations, or parts.

2. **Ignore surrounding textual descriptions** (e.g., title blocks, notes). Focus exclusively on graphical views, symbols, and part depictions.

3. If the drawing contains **multiple sub-figures**, clearly specify which sub-figure each question refers to.

4. Prioritize **clarity, correctness, and answerability**. Avoid vague or ambiguous spatial descriptions.

**Given Drawing Metadata (Provided):**

- **Combination Type:** Single drawing or Multiple sub-figures

- **Drawing Type:** {drawing_type} (provided metadata, used as reference)

- **View Composition:** {view_info} (provided metadata, used as reference)

**Objective:** Identify different views, annotation locations, part locations (for assembly drawings), and cross-view correspondences, and generate precise location-related questions.

**Location Categories to Consider:**

- **View locations**: main view, top view, side view, section view, detail view, auxiliary view, isometric view, exploded view.

- **Annotation locations**: section lines, datum symbols, datum-related tolerances, surface roughness symbols.

- **Part locations (assembly drawings only)**: positions of numbered or named components.

- **Cross-view localization**: correspondence of the same feature across different views; parent locations of detail or auxiliary views.

**Procedure:**

1. Inspect the drawing and identify spatial or positional information from the categories above.

2. Generate **five** professional questions covering different location categories when possible.

3. Each question must explicitly include:

   - the **location type** (view / annotation / part / cross-view),
   - the **target object** (specific view, symbol, part, or feature) and a clear **location description**.

4. If a certain category is not present in the drawing, generate questions from other applicable categories.

**Output Format:**

```
{
  "locations": [
    {
      "type": "location type (view / annotation / part / cross-view)",
      "object": "target object",
      "location_desc": "spatial description",
      "question": "generated professional question"
    }
  ]
}
```

*Figure 12.* Question generation prompt for MechVQA with ground truth: Location

**Role:** You are a professional mechanical engineer with solid expertise in interpreting engineering drawings and performing dimensional reasoning.

**Task:** Carefully analyze the given mechanical drawing and generate high-quality questions that require *dimensional calculation*. The target dimensions are **not explicitly annotated** in the drawing but can be obtained through *linear calculations* based on existing dimensions.

**Key Instructions:**

1. Generate questions **only for dimensions that are not directly labeled** in the drawing.

2. The target dimension must be computable using **linear operations only** (addition or subtraction) from explicitly annotated dimensions.

3. **Do not generate questions** that require complex geometric reasoning (e.g., trigonometry, Pythagorean theorem).

4. Clearly specify the **start and end locations** of the dimension to be calculated. Ignore surrounding textual descriptions and focus only on graphical content and dimension annotations.

**Given Drawing Metadata (Provided):**

- **Combination Type:** Single drawing or Multiple sub-figures

- **Drawing Type:** {drawing_type} (provided metadata, used as reference)

- **View Composition:** {view_info} (provided metadata, used as reference)

**Objective:** Identify locations or features whose dimensions are not directly annotated but can be computed from other labeled dimensions, and generate precise calculation-oriented questions.
**Procedure:**

1. Identify up to **five locations/features** that satisfy all of the following:
   - the dimension is **not explicitly annotated**;
   - the value can be obtained through **linear calculation** (addition or subtraction);
   - the required referenced dimensions are clearly annotated in the drawing.

2. For each identified location, generate a professional question that includes:
   - the **calculation target** (explicit start and end points);
   - the **view** in which the calculation is performed (e.g., front view, top view, section view);the **calculation basis** (which annotated dimensions are used).

3. If fewer than five valid locations exist, generate questions for as many as are reasonably supported.

**Question Examples:**

- "In the front view, what is the distance from the left end face to the center of the middle hole?"

- "In the top view, what is the spacing between the two bosses?"

**Output Format:**

```
{
  "calculations": [
    {
      "location": "description of the start and end positions",
      "view": "corresponding view",
      "question": "generated calculation question"
    }
  ]
}
```

*Figure 13.* Question generation prompt for MechVQA with ground truth: Calculation

## B.5. Validation and Fixing Prompt for MechVQA

**Role.** You are an engineer proficient in mechanical drawing standards. Your task is *not* to answer the question, but to verify whether it is consistent with the drawing and answerable from it.

**Inputs.** A mechanical part / assembly / composite drawing (image); A single-sentence question referring to the drawing; The question's difficulty level and subcategory.

**Principles.** Reason strictly based on the *actual drawing content*:

- If text contradicts the drawing, the drawing prevails;

- For composite drawings, inspect only the referenced subfigure;Do not assume unseen views, annotations, or dimensions.

**Step 1: Read the Drawing.**

- Identify principal views (front, top, side) using standard projection alignment;

- Identify special views: section, auxiliary, enlarged, exploded, and isometric views;

- Distinguish:Auxiliary views (arrow + letter, with a corresponding view);Independent section views (cutting line and/or removed axial sections);

- Identify cutting lines, section labels, datum symbols, dimensions, tolerances, surface roughness, fits, and technical notes.

**Step 2: Check Hidden Assumptions.** Common errors include:

- Referring to nonexistent symbols, views, annotations, or dimensions;

- Confusing datum symbols with section identifiers; Treating local section regions as independent section views;

- Ambiguous language that cannot uniquely identify a feature or dimension.

**Step 3: Verdict.** Select exactly one:

- `correct`: consistent and uniquely answerable;

- `ill-posed`: factually incorrect or unanswerable;

- `ambiguous`: partially supported but with multiple interpretations.

**Step 4: Problem Type Tags.** Choose one or more:

- `nonexistent_symbol`, `wrong_view_reference`, `missing_dimension`, `datum_misinterpreted_as_section`, `misread_tolerance`, `other`.

**Step 5: Fixed Question.**

- If `ill-posed` but fixable, provide a minimally revised valid question;

- If unfixable, output `null`;If `correct` or `ambiguous`, output `null` unless clarification is helpful.

**Output Format (JSON only).**

```
{
    "verdict": "correct | ill-posed | ambiguous",
    "problem_types": [...],
    "explanation_zh": "...",
    "fixed_query_zh": "..."
}
```

*Figure 14.* Prompt for question validation and fixing

**Role.** You are an expert evaluator responsible for voting, merging, and refining multiple candidate answers. Your goal is to produce a single, high-quality final answer while strictly enforcing language consistency between the question and the answer.

**Inputs.** A question; A capability type indicating the evaluation dimension; A list of 2–3 candidate answers; The target language of the question (Chinese or English).

**Primary Constraint: Language Purity.** Language consistency has the highest priority:

- The final answer must strictly use the specified language, mixed-language expressions are not allowed;

- Non-essential foreign words must be translated; Technical terms must use standard translations in the target language.

**Step 0: Question Language Check (Priority).**

- Verify whether the question strictly matches the specified language and detect mixed-language expressions;

- If needed, provide a corrected version of the question with unified language; Record detected issues and whether correction was applied.

**Step 1: Analyze Candidate Answers.**

- Identify the core content of each answer and roup answers with equivalent semantics but different wording;

- For identification questions, check consistency of key facts (e.g., values, names); For reasoning or judgment questions, compare conclusions and main arguments.

**Step 2: Voting and Selection.**

- Count the support for each semantic group;

- Select the group with the highest vote count as the winner; Resolve conflicts by favoring logical soundness and majority support.

**Step 3: Answer Merging.**

- Merge accurate and complementary information from the winning group;

- Remove redundancy while preserving completeness and professionalism; Ensure the merged answer strictly follows the target language.

**Step 4: Final Language Check.**

- Verify that the final answer contains no mixed-language expressions;

- Ensure all technical terms are correctly expressed in the target language.

**Output Format (JSON only).**

```
{
  "question_check": {
    "has_mixed_language": true/false,
    "needs_correction": true/false,
    "issues": [...],
    "corrected_question": "..."
  },
  "answer_groups": [
    {
      "answer_indices": [...],
      "vote_count": N
    }
  ],
  "winner_group": 1,
  "final_answer": "...",
}
```

*Figure 15.* Prompt for answer voting and merging with language consistency

### B.6. Dataset Split

We construct the train, validation, and test splits with an 8:1:1 ratio at the drawing-group level. All QA pairs derived from the same drawing are kept in the same split, so the final QA-pair counts follow the drawing-level allocation rather than being independently sampled per question. To verify that our split is distributionally consistent, we visualize the per-sample representations with t-SNE. As shown in Figure 16, the train, validation, and test sets exhibit similar coverage and cluster structure in the embedding space, without an obvious split-specific shift or missing clusters. This suggests that the three splits are relatively well matched, supporting fair evaluation of generalization.

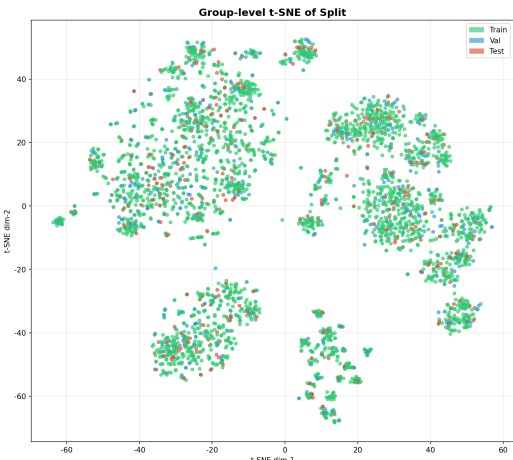

*Figure 16.* t-SNE visualization of the feature distributions for the train/validation/test splits. Different colors indicate samples from each split.

## C. Training Details

All experiments were conducted on a computing cluster equipped with eight NVIDIA H800 GPUs (80GB memory each). Our training pipeline consists of multiple stages: supervised fine-tuning (SFT) followed by reinforcement learning (RL).

### C.1. Supervised Fine-Tuning

In the SFT stage, the model was trained using a standard cross-entropy loss over high-quality question–answer pairs to obtain a stable initialization for subsequent RL optimization.

**Input Processing.** Input images were resized while preserving the original aspect ratio, with the longer edge constrained to 1,024 pixels and the total pixel count capped at 262,144. The input sequence was constructed by concatenating a system prompt, the question (including textual context cropped from the drawing), and the ground-truth answer. The maximum sequence length was set to 4,096 tokens.

**Optimization.** We fine-tuned all model parameters for three epochs using the AdamW optimizer with a learning rate of $1.0 \times 10^{-5}$, a weight decay of $0.01$, and a global batch size of 64. A cosine learning rate schedule with a warmup ratio of $0.1$ was applied. To improve memory efficiency, DeepSpeed ZeRO-3 was employed during training.

### C.2. Reinforcement Learning

Following SFT, we further optimized the model using reinforcement learning. We evaluated three algorithms: Group Relative Policy Optimization (GRPO), Group Soft Policy Optimization (GSPO), and Decoupled Clip and Dynamic Sampling Policy Optimization (DAPO). All methods shared a unified data processing pipeline, optimizer configuration, and distributed training infrastructure, as summarized in Table 6.

**Common Settings.** All RL experiments used the AdamW optimizer with a base learning rate of $1.0 \times 10^{-6}$ and a global batch size of 128. A linear learning rate schedule was adopted throughout RL training. To reduce GPU memory consumption,

optimizer states and the reference model were offloaded to CPU memory under the FSDP framework.

**GRPO Configuration.** For GRPO, we sampled 10 outputs per input to compute group-relative advantages. A low-variance KL penalty with weight $\beta = 0.01$ and a clipping parameter $\epsilon = 0.2$ were applied. Unless otherwise specified, GRPO training was performed for one epoch.

**GSPO and DAPO Configuration.** Both GSPO and DAPO disabled the KL penalty. GSPO employed a sequence-level averaging and a narrow clipping range between $3 \times 10^{-4}$ and $4 \times 10^{-4}$. DAPO adopted a wider clipping range of $[0.20, 0.28]$ and enabled online filtering to dynamically refine training samples during optimization.

Figure 17 summarizes the validation accuracy and response length trends under different RL algorithms.

| Category | Hyperparameter | SFT | RL | | |
| --- | --- | --- | --- | --- | --- |
| | | | **GRPO** | **GSPO** | **DAPO** |
| Data | Max Pixels | 262,144 | 262,144 | 262,144 | 262,144 |
| | Max Sequence Length | 4,096 | 1,024 / 2,048 | 1,024 / 2,048 | 1,024 / 2,048 |
| Optimization | Optimizer | AdamW | AdamW | AdamW | AdamW |
| | Learning Rate | $1.0 \times 10^{-5}$ | $1.0 \times 10^{-6}$ | $1.0 \times 10^{-6}$ | $1.0 \times 10^{-6}$ |
| | LR Scheduler | Cosine | Linear | Linear | Linear |
| | Warmup Ratio | 0.1 | 0.0 | 0.0 | 0.0 |
| | Weight Decay | 0.01 | 0.01 | 0.01 | 0.01 |
| | Global Batch Size | 64 | 128 | 128 | 128 |
| Algorithm | KL Coefficient ($\beta$) | – | 0.01 | Disabled | Disabled |
| | KL Penalty Type | – | Low-var | – | – |
| | Group Size(Sampling) | – | 10 | 10 | 10 |
| | Loss Averaging | – | Token | Sequence | Token |
| | Clip Ratio (Low / High) | – | 0.20 / 0.30 | $3 \times 10^{-4}$ / $4 \times 10^{-4}$ | 0.20 / 0.28 |
| | Online Filtering | – | False | False | True |
| Infrastructure | Hardware | | $8 \times$ NVIDIA H800 (80GB) | | |

*Table 6.* Training hyperparameters for supervised fine-tuning (SFT) and reinforcement learning (RL).

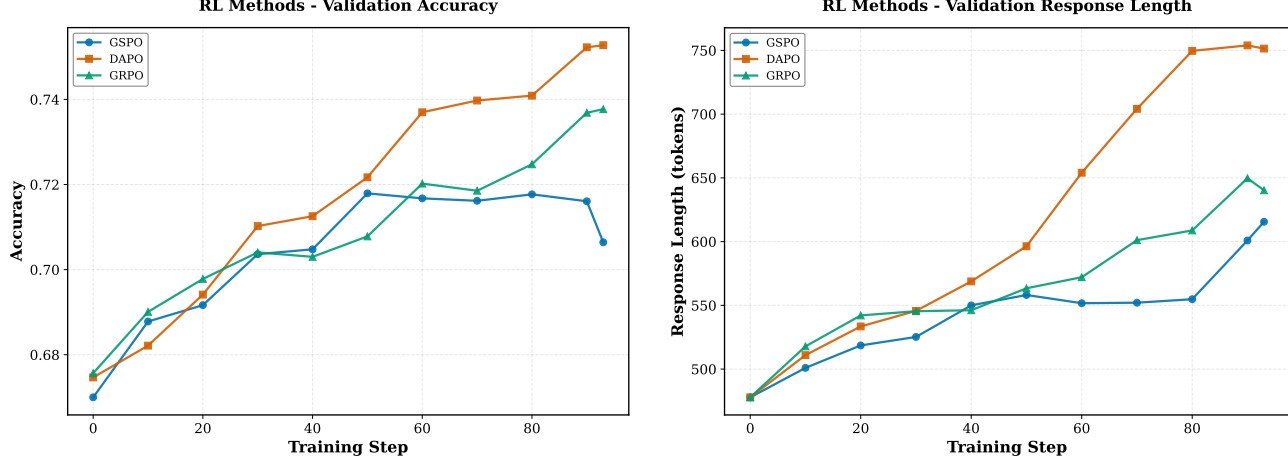

*(a)* Validation accuracy increases during RL training.  *(b)* Validation response length increases during RL training.

*Figure 17.* Training dynamics across different RL algorithms: as training progresses, the model produces longer responses and achieves higher validation accuracy.

## C.3. Reward-Weight Ablation

Table 7 reports reward-weight ablations with the format reward weight fixed at 0.10. The final 0.60/0.30/0.10 accuracy/quality/format weighting achieves the best overall voted score.

| Accuracy weight | Quality weight | Format weight | Total |
|---|---|---|---|
| 0.75 | 0.15 | 0.10 | 83.23 |
| 0.45 | 0.45 | 0.10 | 83.58 |
| **0.60** | **0.30** | **0.10** | **84.85** |

*Table 7.* Reward-weight ablation in targeted RL.

## C.4. Automatic Evaluation Protocol

For benchmark evaluation, we first extract the content inside the model's `<answer>` tag when such a tag is present; otherwise we use the text after `</think>` or the full response as the final answer. Each final answer is then evaluated independently by three LLM judges, GPT-OSS-120B, DeepSeek-V3.2, and Kimi-k2, with temperature set to 0.1. The judges receive only the question, the ground-truth answer, and the model answer, not the original image or the target model identity. This makes the automatic evaluation a post-hoc answer verification step rather than a second attempt to solve the visual problem.

---

Please evaluate whether the following VQA answer is correct.
**Question:** {question}
**Question type:** VQA
**Correct answer:** {correct_answer}
**Model answer:** {model_answer}
**Scoring rules.** Evaluate semantic consistency between the model answer and the correct answer. Surface wording may differ, but the core meaning and key information must match. Assign a score of 1 if the model answer is correct or basically correct and contains all key information. Assign a score of 0 if the model answer is wrong or irrelevant. The score must be binary, either 0 or 1.
**Output format.** Return only a JSON object:

```
{"score": <0 or 1>, "reason": "brief explanation"}
```

---

*Figure 18.* Prompt template for automatic VQA answer evaluation, translated from the implementation.

The returned JSON is parsed by extracting the first JSON object in the judge response; the score is clipped to $[0, 1]$. If JSON parsing fails, the evaluator falls back to a conservative string-level parse for explicit 0/1 outputs, and otherwise assigns 0. For each model answer, we aggregate the valid judge scores by rounding each score to one decimal place and selecting the most frequent score among the three judges. If a model produces no answer, its score is 0. If a judge call fails, that judge result is marked invalid and excluded from the vote; if all judge calls fail, the final score is 0.

