# OpenReview forum: "MechVQA: Benchmarking and Enhancing Multimodal LLMs on Comprehensive Mechanical Drawing Understanding"
_ICML.cc/2026/Conference — ICML 2026 regular_

### Official Review · Reviewer_UfAs · 2026-02-26

**Soundness:** 3
**Presentation:** 2
**Significance:** 3
**Originality:** 2
**Overall Recommendation:** 3
**Confidence:** 3

**Summary:**

This paper introduces MechVQA, a dataset of mechanical drawings with 21k QA pairs over 3.3k images, organized into Recognition/Reasoning/Judging abilities and 10 sub-tasks. It further proposes MechVL, trained via multi-stage procedures (including SFT and self-play RL), establishing a strong open baseline that reportedly outperforms several existing multimodal LLMs on the benchmark. The work aims to accelerate progress on domain-specific, rule-heavy technical diagram understanding.

**Compliance With Llm Reviewing Policy:**

Affirmed.

**Final Justification:**

No rebuttal. I will maintain my original assessment.

**Key Questions For Authors:**

1. How do you ensure QA unambiguity and single-ground-truth correctness for each sub-task? Any inter-annotator agreement statistics?
2. What contamination checks were performed to ensure no training/eval overlap?
3. Provide sub-task-level results with CI/significance and error-type analysis (OCR vs projection vs reasoning).
4. What are the dataset’s licensing terms and how can researchers access it?

**Limitations:**

Partially. Please add explicit limitations regarding domain shift to real industrial drawings, dependence on OCR quality, and constraints on public release/licensing. Include contamination risks and mitigation.

**Strengths And Weaknesses:**

Strengths
1. High practical relevance: mechanical drawings are structured, rule-based, and challenging for general-purpose MLLMs.
2. Clear benchmark framing: multi-skill taxonomy and sub-task coverage facilitate diagnostic evaluation.
3. Provides a strong baseline model and training recipe that can anchor future work.

Weaknesses
1. Ambiguity control is critical for technical drawings; the paper should better document how questions ensure unique, verifiable answers.
2. Potential data contamination: stronger evidence is needed that training data does not overlap with evaluation data (especially if auto-generation or web sources are involved).
3. Need per-subtask breakdown and statistical analysis (CI/significance) to support “overall score” claims robustly.
4. Release/licensing of drawings (often from manuals/textbooks) needs explicit clarification.

---

### Official Review · Reviewer_oBJ5 · 2026-03-10

**Soundness:** 3
**Presentation:** 3
**Significance:** 2
**Originality:** 2
**Overall Recommendation:** 3
**Confidence:** 3

**Summary:**

This paper aims to investigate the MLLMs ability in complex mechanical engineering drawings.  To address this, the paper introduces MechVQA, a comprehensive new benchmark containing 3.3K real drawings and 21K QA pairs that test recognition, reasoning, and judging capabilities. Furthermore, they develop MechVL, a training pipeline via SFT and RL. Experimental results show that the model trained by MechVL significant outperforms other open/proprietary models.

**Compliance With Llm Reviewing Policy:**

Affirmed.

**Key Questions For Authors:**

No question

**Limitations:**

Yes, the author discussed the limitation.

**Strengths And Weaknesses:**

**Strengths**
1. The proposed dataset is useful for some audience.
2. The paper is well structured and easy to follow

**Weaknesses**
1. The proposed training procedure is trivial and bored, which is the defacto sft+rl pipeline. I think can not consider it as a contribution.
2. The comparison is not fair, baseline models are "off-the-shelf" without fine-tuned on the domain-specific data.

---

### Official Review · Reviewer_iURg · 2026-03-13

**Soundness:** 3
**Presentation:** 3
**Significance:** 2
**Originality:** 2
**Overall Recommendation:** 4
**Confidence:** 2

**Summary:**

This work introduces MechVQA and MechVL to advance multimodal understanding of mechanical engineering drawings, a domain that challenges general-purpose MLLMs due to dense annotations, domain-specific symbols, strict projection rules, and the need for cross-view and constraint-aware reasoning. MechVQA is presented as the first comprehensive benchmark for this setting, containing 3.3K real mechanical drawings and 21K question–answer pairs spanning 10 fine-grained tasks across recognition, reasoning, and judgment capabilities, with additional difficulty stratification for systematic evaluation. Built on this benchmark, MechVL is a domain-specialized model trained through a multi-stage paradigm combining supervised fine-tuning and self-play reinforcement learning to improve reliability on drawing-centric tasks. Experimental results show that MechVL outperforms strong general and closed-source baselines on MechVQA.

**Compliance With Llm Reviewing Policy:**

Affirmed.

**Key Questions For Authors:**

Based on the Section 3.3, I wonder if you could provide number that without expert audit, how the performance would be?

Why I ask this is due to I wonder if the method could be extended to a scalable data annotation pipeline. With involving human, it is hard to be scalable.

**Limitations:**

Plz check the points raised above.

**Strengths And Weaknesses:**

Tackling with specific missing pieces for VLM is great contribution to our community and the proposed Mechanical Drawing dataset is one of them. All the used technical components sound. The paper is easy to follow with clean figures. They also conduct reasonably comprehensive experiments with comparing lots of VLMs and ablating many components.

The proposed data annotation pipeline is hard to scale. On the top of it, the proposed dataset is also limited at scale. Besides, I mostly feel this paper is more like a technical report instead of conference paper due to limited novelty.

---

### Official Review · Reviewer_f532 · 2026-03-16

**Soundness:** 3
**Presentation:** 3
**Significance:** 3
**Originality:** 3
**Overall Recommendation:** 5
**Confidence:** 4

**Summary:**

MechVQA introduces the first comprehensive VQA dataset for mechanical engineering drawings: 3.3K images, 21K QA pairs across 10 subtasks organized into Recognition, Reasoning, and Judging categories. The paper also proposes MechVL, a training pipeline that applies supervised fine-tuning (SFT) on Qwen3-VL-4B-Instruct followed by self-play reinforcement learning using DAPO (Direct Alignment from Preference Optimization). A composite reward function balances accuracy, format compliance, and answer quality. MechVL-4B-RL achieves 84.85 total score, outperforming the best open-source baseline by +5.94 and the best closed-source baseline by +7.57.

**Compliance With Llm Reviewing Policy:**

Affirmed.

**Key Questions For Authors:**

1. What is the inter-annotator agreement across the 10 subtasks, and who performed the annotations (students, engineers, or domain experts)?
2. Are the baselines (open and closed-source) evaluated zero-shot or fine-tuned on MechVQA training data? If zero-shot, can you include fine-tuned baselines?
3. What is human expert performance on this benchmark?
4. How diverse are the drawing sources — textbook, industry, synthetic? What ISO/ASME standards do they follow?
5. Can you provide an ablation of the composite reward weights and evidence against reward hacking during RL?

**Limitations:**

**Scale is modest.**

3.3K images and 21K QA pairs is relatively small, especially for a benchmark paper. Compare to VQAv2 (~1.1M QA pairs), GQA (~22M), or even specialized benchmarks like ChartQA (~33K). At this scale, benchmark saturation is a risk, and the diversity of mechanical drawings may be limited.

**Annotation quality and inter-annotator agreement.**

Mechanical drawings require domain expertise to annotate correctly. The paper should report: who annotated, their qualifications, inter-annotator agreement metrics (Cohen's κ or similar), and the annotation protocol. Without this, benchmark reliability is uncertain.

**Drawing source diversity.**

Are the 3.3K images from textbooks, industry, generated synthetically, or a mix? Textbook drawings are typically cleaner and more standardized than real-world manufacturing drawings. If the benchmark is textbook-heavy, it may overstate model capabilities on practical applications.

**Limited to 2D drawings?**

Mechanical engineering increasingly uses 3D CAD models. The paper should discuss scope limitations and whether the task taxonomy extends to 3D representations.

**Strengths And Weaknesses:**

1. First large-scale VQA benchmark specifically for mechanical engineering drawings
2. 10-subtask taxonomy spanning Recognition/Reasoning/Judging
3. MechVL training pipeline: SFT + self-play RL with composite reward
4. Strong empirical results against both open and closed-source baselines

---

### Decision · Program_Chairs · 2026-04-30

**Decision:**

Accept (regular)

**Comment:**

This paper received mixed reviews.

All reviewers agree that the benchmark is a valuable contribution, while at the same time raised a few issues for the authors to respond. Most of the issues raised are requests for clarifications, along with a couple of requests that would require new experiments (human performance analysis, extra ablation studies). Methodologically, the pipeline of SFT+RL is not novel, but I understand it as a baseline method, more than a methodological contribution.

It is curious that the paper does not include in its state of the art document VQA benchmarks as mechanical drawings are a specific document type.

This paper was originally desk rejected, a decision that was later reversed by the program chairs. Due to the late desk rejection reversal, the authors were not able to provide a rebuttal. The PCs and AC encouraged the authors to send a rebuttal through the AC-author confidential comment mechanism, but they did not take advantage of this opportunity. As such, the requests for clarification posted by the reviewers remained unanswered.

The issues raised by the reviewers are mostly requests for clarifications and extra information. Although it is true that the paper would benefit from an extra round of improvements, there is no critical technical issue raised that would prohibit the paper from being published. I consider that if rebuttal was possible, a large number of these clarifications would be easy to provide.